# Calibration of Wideband LFM Radars Based on Sliding Window Algorithm

**Hyungwoo Kim** [1], **Jinwoo Kim** [1], **Jinha Kim** [1], **Jaeyoung Choi** [1], **Sangpyo Hong** [2], **Nammoon Kim** [2] **and Byungkwan Kim** [1,*]

1   Radio and Information Communications Engineering, Chungnam National University, Daejeon 34134, Republic of Korea; hyeongwoo.kim@o.cnu.ac.kr (H.K.); jinwoo.kim@o.cnu.ac.kr (J.K.); jinha.kim@o.cnu.ac.kr (J.K.); jaeyoung.choi@o.cnu.ac.kr (J.C.)
2   Hanwha System Co., Ltd., Yong-in 17121, Republic of Korea; sp1110.hong@hanwha.com (S.H.); nammoon.kim@hanwha.com (N.K.)
*   Correspondence: byungkwan.kim@cnu.ac.kr; Tel.: +82-042-821-6883

**Abstract:** This paper addresses the challenges of wideband signal beamforming in radar systems and proposes a new calibration method. Due to operating conditions, the frequency-dependent characteristics of the system can be changed, and the amplitude, phase, and time delay error can be generated. The proposed method is based on the concept of the sliding window algorithm for linear frequency modulated (LFM) signals. To calibrate the frequency-dependent errors from the transceiver and the time delay error from the true time delay elements, the proposed method utilizes the characteristic of the LFM signal. The LFM signal changes its frequency linearly with time, and the frequency domain characteristics of the hardware are presented in time. Therefore, by applying a matched filter to a part of the LFM signal, the frequency-dependent characteristics can be monitored and calibrated. The proposed method is compared with the conventional matched filter-based calibration results and verified by simulation results and beampatterns. Since the proposed method utilizes LFM signal as the calibration tone, the proposed method can be applied to any beamforming systems, not limited to LFM radars.

**Keywords:** phased array radar; radar calibration; wideband signal; true time delay; beamforming





## 1. Introduction

Radar performs various operations like surveillance, imaging, and target detection tracking in a remote location. To perform these operations with high resolution and accuracy, the radar system requires a wideband signal and large number of transmit and receive modules (TRMs) for beamforming [1]. The beamforming can be performed in the time domain by the true-time delay element (TTD) or in the frequency domain by a phase shifter. The phase shifter-based beamforming, which is frequency-domain beamforming, is widely used in radar systems because of its lower cost compared to TTD elements. However, the frequency-domain beamforming has a beam squint problem, which is that the beam direction changes with frequency [2]. The squint angle can be calculated by the following equation:

$$\Delta\theta = sin^{-1}(\frac{f}{f_c} \cdot sin(\theta_0)) - \theta_0 \tag{1}$$

where $f$ is the frequency of the signal, $f_c$ is the center frequency of the signal, $\theta$ is the squinted beam direction, and $\theta_0$ is the desired beam direction.

If the wideband signal is beamformed with the phase shifter, the beam is directed to the undesired direction at other frequencies, and the effective beam width is broadened. Therefore, the time domain beamforming is required for the precise wideband beamforming radar systems [2]. The beamforming using time delay elements can solve the beam squint

problem, but it requires precise control of the time delay between TRMs. The required time delay for the beamforming is proportional to the distance between the phase center of antenna elements. For example, for an S-band radar system with center frequency of 3.25 GHz, the minimum delay between antenna is equivalent to 130 ps. Therefore, precise control of the time delay between TRMs is required, and true time delay elements are adopted for those requirements.

Recent phased array radar systems are designed to operate for multiple functions with various waveforms. Based on the target or operation mode, the radar system selects its waveform, including bandwidth or additional binary phase modulation [3]. This means that the radar system should be able to control its phase using the phase shifter, even if it performs beamforming in the time domain with TTD elements. On the other hand, the radar system can apply both time delay and phase shift for the beamforming by considering the total cost and performance of the system [4].

The TRMs are composed of various components, such as a power amplifier, phase shifter, and many passive elements that have frequency-dependent characteristics [5]. In the narrowband assumption, the frequency-dependent characteristics are negligible, and the TRMs are assumed to have constant characteristics in the frequency domain [6]. However, for a wideband signal such that the fractional bandwidth is more than 10%, the characteristics of the TRM are not constant in the frequency domain, and the characteristics of the TRMs are different from each other [7].

Therefore, the beamforming system requires precise control of the transmitted or received signal to have desired differences in time, amplitude, phase in all frequencies. To achieve this, many calibration methods have been proposed and implemented [8–14]. The PN gating method is a widely used technique for the calibration of the remote sensing radar system, which is based on the orthogonal code [15–17]. The calibration method can monitor characteristics of each TRM by applying the orthogonality of the waveform during the calibration process. This method is simple and easy to implement, and can measure realistic characteristics of the TRMs since it monitors the characteristics of the TRMs when all TRMs are operating simultaneously. However, the PN gating method is still used in the narrowband system, and the calibration is performed with a single value of amplitude, phase, and delay [18].

In this paper, we propose a monitoring and calibration method for the wideband phased array radar system by adopting the sliding window algorithm. The proposed method can monitor the characteristics of the TRMs both in the frequency domain, by the amplitude and phase of the TRMs, and in the time domain, by the time delay of the TRMs. This method does not alter any hardware of the radar system and can be applied to the existing radar system by the proposed processing method. The proposed method can be applied to the radar system with both TTD and the phase shifter-based beamforming system.

The contribution of the proposed method in LFM beamforming radar calibration can be summarized as follows.

- The characteristics of the TRMs, amplitude, phase, and time delay can be monitored and calibrated simultaneously.
- The proposed method can be applied to the LFM radar system with both TTD and the phase shifter based beamforming system.
- The proposed method can be directly applied to the existing radar system without any hardware modification.
- The proposed method is compatible with orthogonal code calibration methods.
- The proposed method does not require additional measurements for the calibration.

The rest of this paper is organized as follows. In Section 2, the principles of the calibration of wideband phased array radar system with linear frequency modulation are described. In Section 3, the proposed error monitoring and calibration method is presented. In Section 4, the simulation setup, parameter, and results are presented. Finally, the conclusion is given in Section 5.

## 2. Calibration of Beamforming System with Linear Frequency Modulation

The linear frequency modulation (LFM) is a widely used waveform for the radar system to achieve high range resolution. The LFM waveform is defined as follows:

$$x(t) = e^{j2\pi(f_0 t + kt^2)} \tag{2}$$

where $f_0$ is the center frequency of the LFM waveform and $k$ is the chirp rate of the LFM waveform.

The matched filter output represents the relative delay, amplitude attenuation, and phase shift of the received signal compared to the reference signal [18]. This information is utilized to find out the range to the target with delay information, or to measure the speed from coherent pulses with constant phase shifts. With the moving platform, a radar can generate the images of the scene with synthetic aperture radar (SAR) processing.

The calibration of the beamforming system with the LFM waveform is started by the transmitting and receiving the calibration signal through calibration path. The calibration path can be configured internally or externally by considering the radar system's structure and complexity [4]. The received calibration signal is performed by matched filtering with the reference signal. The result of the matched filter contains the information of all characteristics of the transmitter, receiver, cables, passive and active components in the calibration path. To remove the effects of the calibration path, the calibration path is usually configured with passive elements with known characteristics.

By calculating the matched filter with a reference signal and calibration signal for all TRMs, the characteristics of the TRMs can be monitored. The maximum value and index of the matched filter output in the digital domain are found for each TRM, which means the calibration signal's amplitude, phase and delay affected by the hardwares of the TRM. Since the beamforming is implemented by the relative difference in the time and frequency domain of the signal for each TRM, the calibration for all TRMs is realized by setting a reference TRM and comparing the relative amplitude and phase of the other TRMs to the reference TRM. Further details and examples of the calibration with the LFM signal, such as the calibration path and processing steps, can be also found in [14,19–23].

The conventional calibration of the beamforming system with LFM waveform is performed by the single value of amplitude, phase, and delay. This means that if the TRM and any component has frequency-dependent characteristics, the calibration result cannot be accurate. To overcome this problem, we propose a new calibration method for a wideband phased array radar system by adopting the sliding window algorithm for frequency-specific calibration.

## 3. Proposed Method

### 3.1. Frequency-Dependent Error Model

As described in the previous section, the conventional calibration can monitor and generate a precise beampattern of the phased array radar system for the narrowband signal. However, if wideband signals are applied, the performance of the beamforming system can be different in frequency for various conditions. The frequency-dependent performance variation of TRMs, phase shifter, and TTDs can be found in the literature [24].

Before describing the proposed method, we define our error model for the beamforming system. Since the radar system considered in this study is based on the LFM waveform, the frequency error can be expressed with the time domain function. The frequency-dependent amplitude and phase function can be represented in the time valued function since the LFM waveform changes its frequency linearly in time:

$$E_N(t) = A_N(t) \cdot exp(j2\pi \cdot \phi_N(t)) \tag{3}$$

where $A_N(t)$ and $\phi_N(t)$ are the amplitude and phase error function of the $N$th TRM of the beamforming system in the frequency domain, respectively. The time variable $t$ can be

converted to frequency variable $f$ by the linear relationship of the LFM signal. On the other hand, the relative delay error for any signal $s(t)$ can be represented as follows:

$$s_{delayed}(t) = s(t - \tau_N) \tag{4}$$

where $\tau_N$ is the relative delay error of the $N$-th TRM of the beamforming system. Therefore, the frequency-dependent error model of the beamforming system can be expressed as follows:

$$x_{N,error}(t) = x_N(t - \tau_N) \cdot E_N(t - \tau_N) \tag{5}$$

where $x_{N,error}(t)$ is the error applied calibration signal, including the time delay error, frequency-dependent amplitude error and phase errors.

In the simulation, the analog time delay error can be implemented by shifting the digital signal by considering the sampling rate. Therefore, the precise delay error requires oversampling and decimation process of the signal.

### 3.2. Calibration with Sliding Window Algorithm

The proposed method is based on the sliding window concept. The conventional calibration performs the matched filter of the received LFM signal to measure the amplitude or phase differences of the calibration signal. The main concept of the proposed method is the matched filtering of the LFM signals from a specific part of the calibration signal by windowing the LFM signal to represent a narrowband characteristic of the system. Figure 1 shows the concept of the proposed method with the sliding window algorithm. The narrowband segmentation of the LFM signal is processed by the matched filter with the same part of the reference signal. By sliding the matched filter window in the time domain, the characteristic of the LFM signal can be monitored in the frequency domain because the LFM signal is linearly changing its frequency.

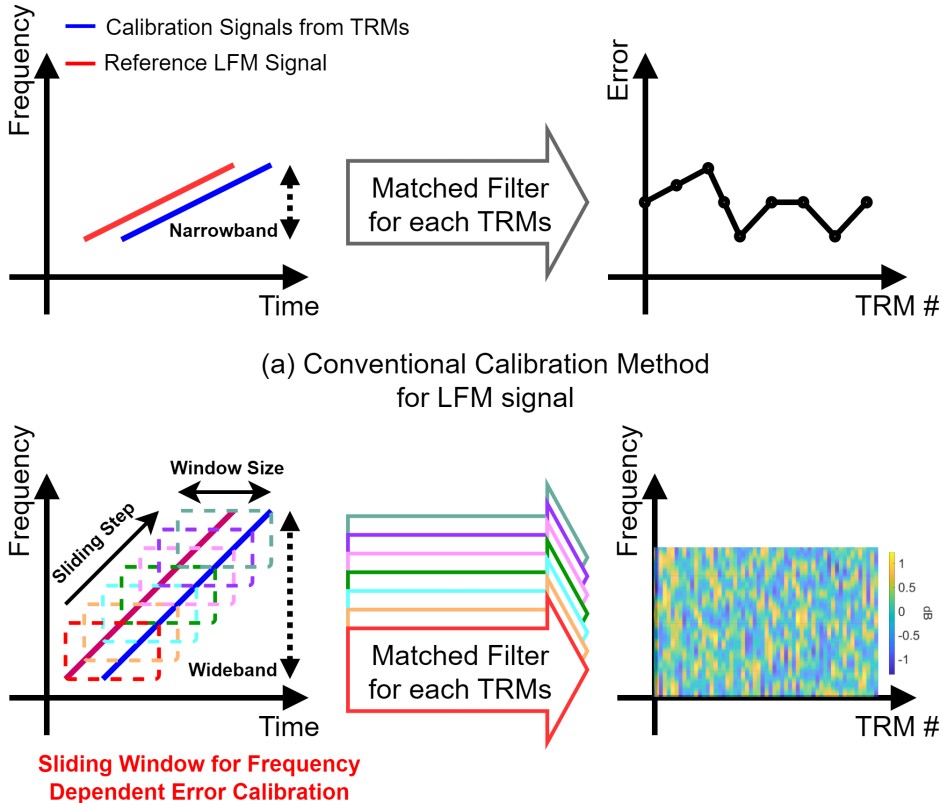

**Figure 1.** The conceptual diagram of the proposed method.

An example system block diagram of the radar system with a calibration path is shown in Figure 2, and the total process of the proposed method is shown in Figure 3. When the calibration process is started, the wideband LFM reference signal is generated. The reference signal is transmitted through the calibration path. To deal with multiple signals for all TRMs, the calibration signal can be obtained by the direct RF method or the orthogonal code like the PN gating method. The direct RF method measures a single calibration signal for one TRM at a time, and repeats the process for all TRMs. On the other hand, the application of orthogonal code to the calibration signal can measure all TRMs at once. After signal separation for all TRMs, the received signal (calibration signal) for each TRM is processed by matched filter with the reference signal.

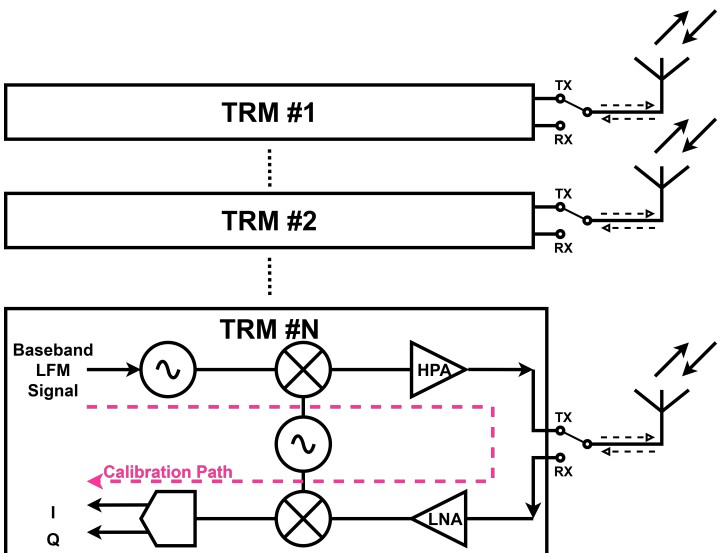

**Figure 2.** LFM radar system block diagram with a calibration path. The calibration path may differ for applications.

A reference TRM is selected, and the coarse delay is estimated by the index difference of the maximum of matched filter output. The coarse delay is compensated by shifting the index of the total signal. After coarse delay compensation, the proposed method is started.

The sliding window algorithm is an algorithm that divides a signal into a certain size of window by applying processing, and moves the window by a certain step size. The window size and step size are the parameters of the sliding window algorithm. The window size is the number of samples for narrowband segmentation of the wideband LFM signal. The step size is the number of samples for shifting the window. The sliding window algorithm is performed to monitor the frequency-dependent characteristics of the TRMs. The amplitude and phase error for a certain frequency can be measured by the maximum value of the matched filter output from a window. Based on the measured error values, the proposed method constructs an error matrix in the frequency domain. The calibration of the TRMs can be performed by applying the error matrix to the received signal.

The calibration of the wideband LFM signal is performed by the same principle of the error model described in the previous section. First, the time delay calibration is implemented by shifting the total signal index by the measured delay error. However, this calibration is not enough to compensate for the precise time delay of the TRMs since the time delay error is represented in the units of sample index of the signal. For example, by supposing 1 GHz bandwidth of the LFM signal with 10 GHz center frequency, the sampling rate of the signal processing unit should be larger than 2 GHz [4]. Assuming that the sampling rate of the analog-digital converter is 2 GHz, then the sampling time of a sample is 0.5 ns, which means that the time delay error can be represented in the units of 0.5 ns. However, the beamforming radar will control the time delay in units of a few picoseconds to generate a high-resolution beampattern, which means that the time delay

error in the units of picoseconds should also be calibrated. Therefore, the index-based delay calibration is named coarse delay calibration, and precise time delay calibration is separately performed in this study. The coarse delay calibration can be enhanced by increasing the sampling rate of the analog-digital converter, but it is not practical in the real system since the beamforming system should process a large amount of the digital signal in real time.

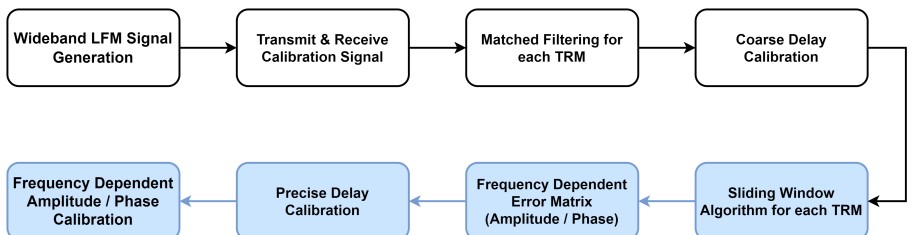

**Figure 3.** The procedure of proposed calibration method. The highlighted block is the proposed method, while normal blocks are conventional methods.

After coarse delay calibration, the signal still contains frequency-dependent amplitude and phase errors, and precise time delay error. The frequency-dependent amplitude and phase errors can be calibrated with the same method of the error model described in the previous section. The phase error in this stage consists of two parts, which are the true frequency-dependent phase error of the TRM and the linear phase term due to precise time delay. Figure 4 illustrates an example of the phase error consisting of two components. Type 2 error involves frequency-dependent amplitude and phase errors, while Type 3 error encompasses frequency-dependent amplitude and phase errors, as well as time delay errors. These errors are elaborated upon in detail in Section 4.3.1.

The calibration of precise time delay can be performed with two different approaches. The first method is the calibration of the true-time delay element with the precise delay error by applying linear regression of the measured phase error. This is effective only when the frequency-dependent phase error is much smaller than the additional linear phase due to the time delay error ($\phi_N << \tau_N * f(t)$).

If the beamforming radar system works fully in the digital domain, the precise time delay can be considered another phase error source that needs calibration. In this situation, the second method can be introduced, the predistortion of the LFM signal with phase error, for the precise time delay and phase error of TRMs. This can be implemented without additional computational cost since the calculation is equivalent to the proposed amplitude and phase calibration.

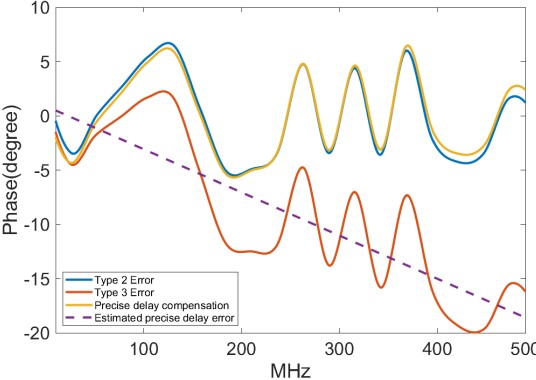

**Figure 4.** An example of frequency dependent phase error is combined with a delay error. Since the delay of LFM signal means a constant frequency difference, the delay error is represented in the linearly increasing phase error. Therefore, the precise delay can be estimated by applying linear regression to the obtained phase error by the proposed method, and can be compensated for by the true time delay elements.

The proposed method can calibrate this phase error and precise time delay simultaneously since both errors are represented in the measured phase error. Therefore, the frequency-dependent amplitude, phase error, and precise time delay of the $N$th TRM can be calibrated by the following equation:

$$x_{calibrated}(t) = x_{N,error}(t) \cdot \frac{1}{A_N(t)} \cdot exp(-j2\pi \cdot \phi_N(t)) \tag{6}$$

## 4. Simulation and Results

### 4.1. Simulation Parameters

To verify the proposed method, the simulation is performed by considering realistic simulation parameters and a beamforming system. The parameters for error models are summarized in the following Table 1. The wideband beamforming radar system, which is based on both the phase shifter and true time delay, is considered in the simulation as presented in [4]. The beamforming system consists of 64 TRMs, and the LFM signal is generated with 0.5 GHz bandwidth, with a 50 us pulse length and 3.25 GHz center frequency.

**Table 1.** The waveform and beamforming system parameters.

| Parameters | Value | Unit |
| --- | --- | --- |
| Central Frequency | 3.25 | GHz |
| Bandwidth | 0.5 | GHz |
| Pulsewidth | 50 | us |
| No. of TRMs | 64 | |
| No. of TTDs | 8 | |

### 4.2. Error Model

The range of random error values are referred from the measured error values of time-delay elements and amplifiers [25]. The frequency dependent amplitude and phase error are modeled with interpolation from random values from the reported typical error range [25]. Each frequency dependent error is randomly generated by the uniform distribution of the error range in Table 2. The errors are assigned to each TRM by a frequency amplitude matrix and frequency phase matrix. The example of error model and matrix will be presented in the following section. The error values are multiplied with the reference LFM signal to simulate the errors of the beamforming system.

**Table 2.** The amplitude, phase, and time error values for the simulation. The error values are selected to simulate the error levels similar to the single-bit range of phase shifter, the true time delay element.

| Parameters | Value | Unit |
| --- | --- | --- |
| Amplitude Error | −1.1∼0 | dB |
| Phase Error | −7∼7 | degree |
| Coarse Time Delay Error Unit | 100 | ps |
| Precise Time Delay Error Unit | 20 | ps |

The time delay error is also referred to as the typical error range of the true time delay elements for the beamforming system [25]. In the digital domain, the time delay can be assigned to a signal by shifting the index of the signal. Therefore, the oversampling and decimation process is introduced to represent the time delay error lower than the sample time interval. The coarse delay error can be represented with 0.1 ns step, and the precise time delay error is represented with a 20 ps step in the simulation.

By applying these error models to the reference LFM signal, the error signal with frequency dependence and time delay can be generated.

*4.3. Scenario and Results*

4.3.1. Error Measurement

In this simulation, we assign errors to the selected TRMs to verify the error measurement performance of the proposed method. To measure the various types of error, we assign the following different error model to the TRMs. Conventional and proposed calibration methods are performed for the error signal, and the error values are compared with the assigned error values.

Three different types of error are prepared for the simulation to verify the proposed method and compare with the conventional method. The first type (Type 1) of error is the frequency constant amplitude and phase error, which is the most common error model for the narrowband beamforming system. The second type (Type 2) of error is the frequency-dependent amplitude and phase errors from the electromagnetic components, and the third type (Type 3) of error is the frequency-dependent amplitude and phase error with time delay error from the true time delay element.

For clear comparison, the error values are assigned to the TRMs as follows. The type 1 error model is assigned to the TRM 10, and the type 2 error model is assigned to the TRM 20, and the type 3 error model is assigned to the TRM 30.

- Frequency constant amplitude and phase error (Type 1)

The amplitude error is assigned as −3 dB, and the phase error is assigned as 5 degrees. The error signal is generated by multiplying the error model to the reference signal, and the amplitude and phase of the error signal are shown in Figure 5.

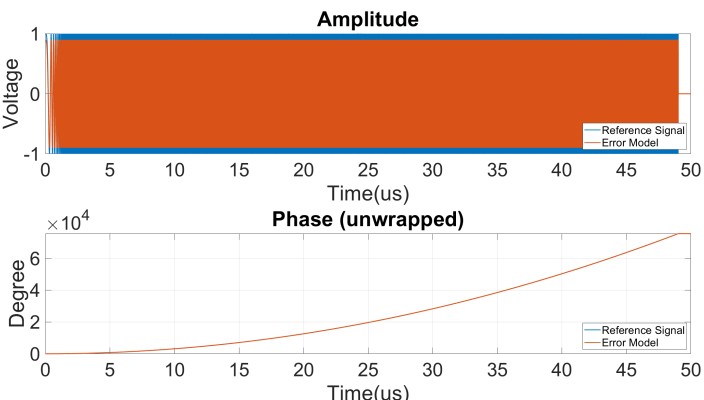

**Figure 5.** The amplitude and phase of frequency constant error applied signal.

After calibration with the conventional and proposed methods, the result error values are compared with the assigned error values for the frequency of LFM signal as shown in Figure 6. Since the conventional method presents the error values with the same value for all frequencies, the error values are presented as a line. Both methods present the error values with the same value for the amplitude and phase, which means that the conventional method can calibrate the LFM signal when the error is constant for all frequencies. In this case, the proposed method can also calibrate the error, but the computational complexity is higher than the conventional methods since it generates same values for all frequencies.

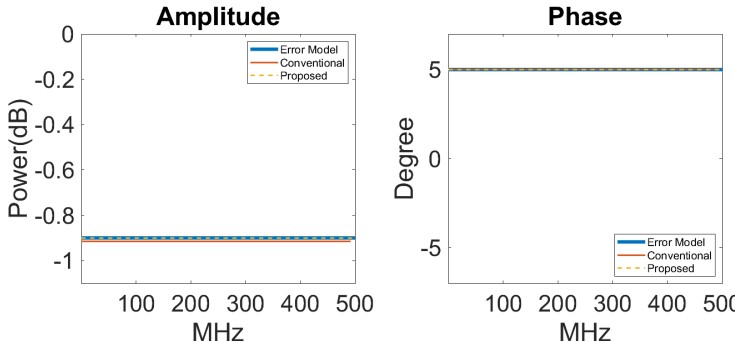

**Figure 6.** The assigned amplitude and phase error are presented with the results from the conventional method and the proposed method.

- Frequency-dependent amplitude and phase error (Type 2)

The frequency-dependent error is assigned as the function of frequency, which is generated by interpolating the random values from the reported typical error range. An example of the error applied signal is shown in Figure 7.

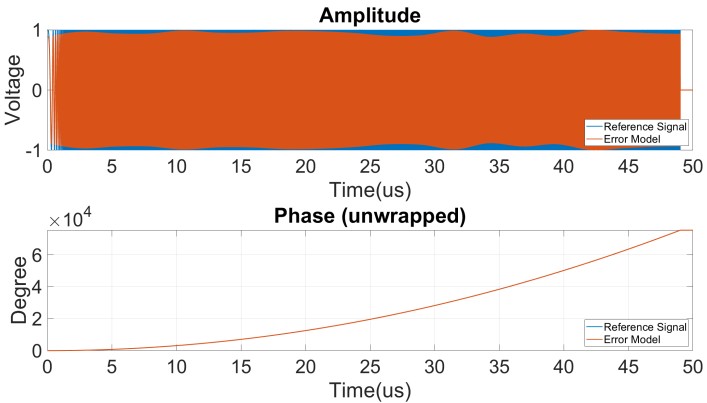

**Figure 7.** The amplitude and phase of frequency-dependent amplitude and phase error applied signal.

The calibration result of error signal presented in Figure 7 is shown in Figure 8. In the same manner, the error values from the conventional calibration are presented as a line because it does not consider frequency dependent error. On the other hand, the proposed method presents the error values for each frequency bin. The calculated calibration accuracy of the amplitude error is 0.016 dB, and the phase error is 3.523 degrees. The proposed method can calibrate the frequency-dependent error, but the selection of the window size and sliding step size is an important factor to the calibration accuracy.

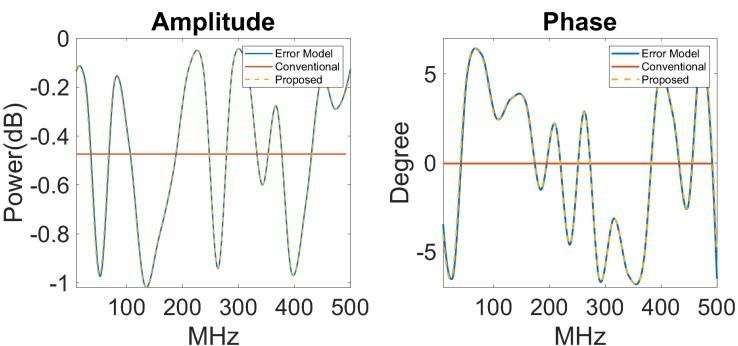

**Figure 8.** The assigned amplitude and phase error are presented with conventional and proposed calibration results for the type 2 error model.

To show the effect of the window size and sliding step size, the calibration results are shown in Figure 9. The window size is set to 500 and 4000, which slices 50 ns and 400 ns length of the LFM signal. The sliding step is determined by the overlapping ratio, which is set to 85% and 30%, 75 and 2800 samples for 500 ns and 4000 ns window sizes, respectively. Both parameters determine the number of error values versus frequency, which is directly related to the calibration accuracy. This is due to the nature of the LFM signal, which has a linear frequency increase in time. The smaller window size will represent the narrowband characteristics, and the larger overlapping ratio will contribute the changing characteristics of the error values in the frequency domain. Therefore, the larger window size and smaller overlapping ratio shows lower calibration accuracy as shown in Figure 9.

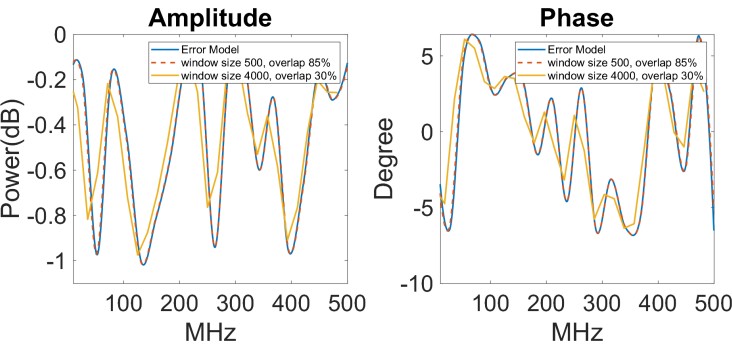

**Figure 9.** The proposed calibration result with different window size and overlapping ratio. The smaller window size and larger overlapping ratio shows higher calibration accuracy. The larger overlapping ratio means a smaller sliding step size.

- Frequency dependent amplitude and phase error with time delay error (Type 3).

As described in the proposed calibration process, the coarse delay error is calibrated first, and the precise delay error is calibrated after the coarse delay error is calibrated. Therefore, the phase error from the precise delay error is included in the error signal as presented in Figure 4, and it needs to be calibrated with the proposed method. In this simulation, the total delay error is set to 100 ps and calibrated coarsely, and the precise delay error is set to 20 ps. The error signal with the precise delay error is shown in Figure 10. Figure 10 shows that the phase difference of the error signal is increasing with time, which is from the precise delay error.

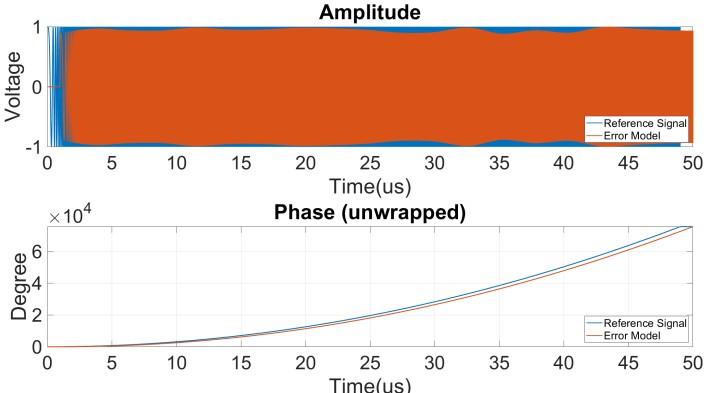

**Figure 10.** The amplitude and phase of frequency-dependent amplitude, phase error and time delay error applied signal example. We applied 1 us delay for better presentation of time delayed signal. Due to the time delay error, the phase difference of the error signal is increasing with time.

The calibration result of error signal with precise time delay is presented in Figure 11.

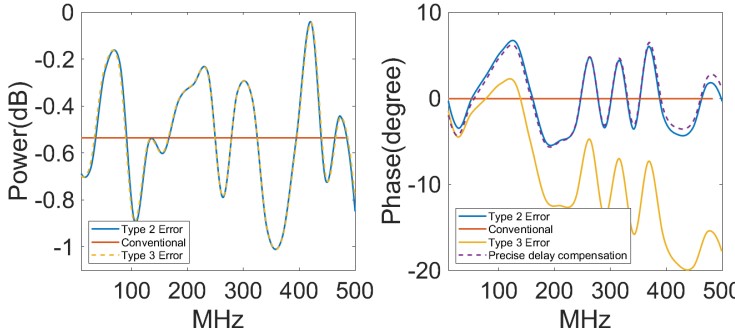

**Figure 11.** The assigned amplitude and phase error are presented with conventional and proposed calibration results for type 3 error model.

From the calibration result, the precise time delay does not affect the calibration accuracy of the amplitude error, but it affects the calibration accuracy of the phase error. As presented in Figure 11, the applied phase error is added with a linear function of frequency. By applying simple linear regression model to the phase error, the slope of the phase error is calculated as −3 degree/GHz, equivalent to 16 ps of precise delay. The difference of the precise delay error applied and obtained is 4 ps, which is below the requirement of the true time delay element resolution.

In summary, the proposed calibration method is compared with the applied error model and the conventional calibration method for three different error types, presented in Figure 12 and Table 3.

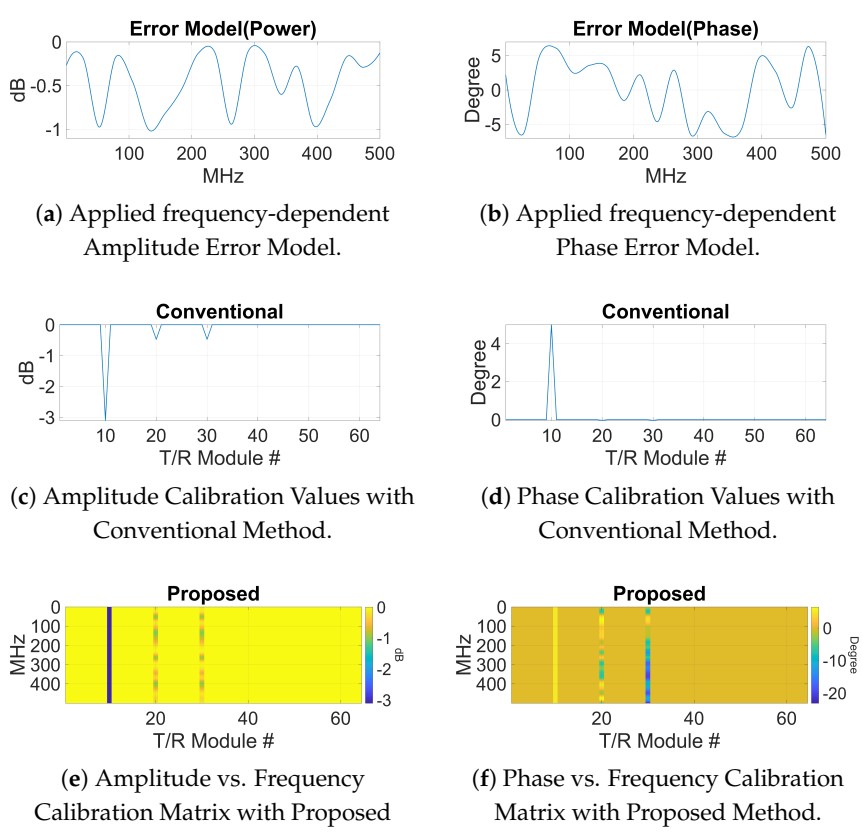

(**a**) Applied frequency-dependent Amplitude Error Model.

(**b**) Applied frequency-dependent Phase Error Model.

(**c**) Amplitude Calibration Values with Conventional Method.

(**d**) Phase Calibration Values with Conventional Method.

(**e**) Amplitude vs. Frequency Calibration Matrix with Proposed Method.

(**f**) Phase vs. Frequency Calibration Matrix with Proposed Method.

**Figure 12.** The TRM numbers 10, 20, and 30 are assigned with the constant amplitude and phase error, frequency-dependent amplitude and phase error, and frequency-dependent amplitude and phase error with precise time delay error, respectively. The conventional and proposed calibration method are performed to the find out errors from the reference LFM signal.

**Table 3.** The calibration result with the conventional and proposed method for three different type of errors. The error is measured with RMSE value for amplitude and phase error, and the time delay error. Note that the time delay error is only available for the type 3 error and proposed method.

| Table | Conventional | | | Proposed | | |
|---|---|---|---|---|---|---|
| | Amplitude (dB) | Phase (Degree) | Time (ps) | Amplitude (dB) | Phase (Degree) | Time (ps) |
| Type 1 Error | 0.0151 | 0 | N/A | 0.0151 | 0 | N/A |
| Type 2 Error | 1.4910 | 3.3972 | N/A | 0.0235 | 0.2753 | N/A |
| Type 3 Error | 3.3972 | 20.9799 | N/A | 0.0235 | 0.6599 | 4 |

Therefore, the proposed calibration method can calibrate the frequency-dependent amplitude and phase error with the precise time delay error. In the next section, the calibration results of the conventional method and proposed method are utilized to generate the beamforming result.

### 4.3.2. Calibration Effects on Beamforming

In this scenario, all TRMs are assigned with randomly generated errors, and the conventional and proposed calibration methods are performed. Figure 13 shows the error matrix from the conventional and proposed calibration methods for all TRMs. The assigned error values are randomly generated as described in Table 2.

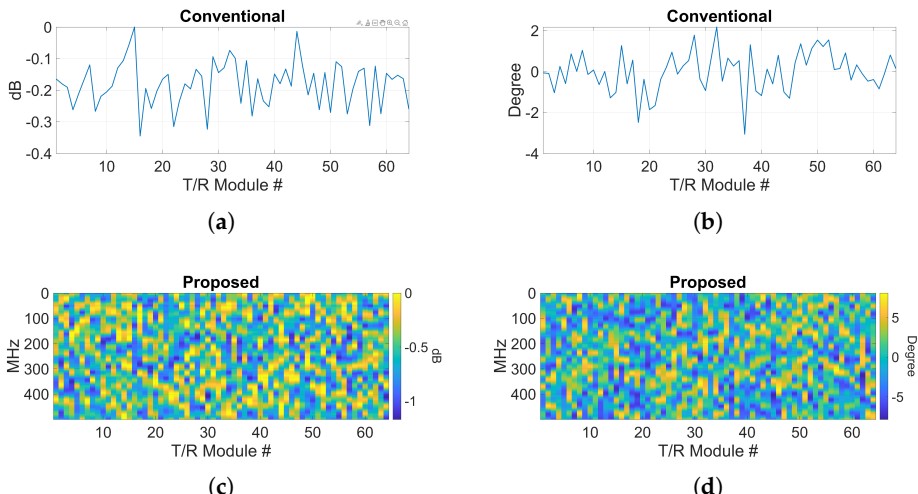

(a)

(b)

(c)

(d)

**Figure 13.** The calibration result with randomly generated errors for all TRMs. The error values are randomly generated as described in Table 2. (**a**) The amplitude errors for each TRM by the conventional calibration method. (**b**) The phase errors for each TRM by the conventional calibration method. (**c**) The amplitude errors in terms of frequency for each TRM by the proposed calibration method. (**d**) The phase errors in terms of frequency for each TRM by the proposed calibration method.

Supposing that the TRMs are performing beamforming in the direction of −35 degree in the azimuth plane, the expected beamforming result is presented in Figure 14a. Based on the measured error value from the conventional method and proposed method, the beamforming result is presented in Figure 14b,c.

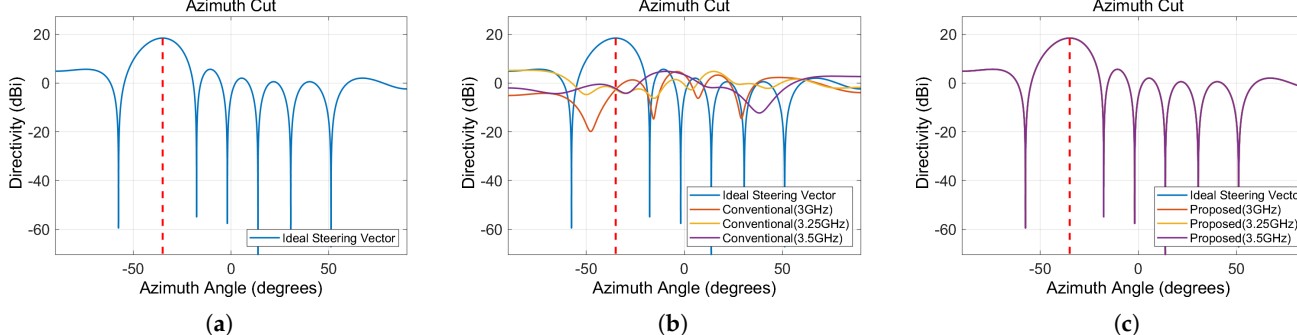

**Figure 14.** (**a**) The desired beamforming result with ideal steering vector. (**b**) The beamforming result with steering vector calibrated with the conventional method. Due to the frequency-dependent amplitude and phase error described in Figure 13, the beamforming result is distorted for all LFM frequencies. (**c**) The beamforming result with steering vector calibrated by the proposed method. Compared to the conventional method, the proposed method can calibrate the frequency-dependent amplitude and phase error with time delay error.

From the beamforming result, the proposed method can calibrate the frequency-dependent amplitude and phase error with the time delay error, and the beamforming result is not distorted for all LFM frequencies. However, the conventional method cannot calibrate the frequency-dependent amplitude and phase error, and the beamforming result is distorted for all LFM frequencies. The main contribution for the beam distortion is the frequency-dependent phase error, which is from the characteristic of the TRM and precise delay error.

### 4.4. Discussions

The proposed method calibrates both precise time delay error and phase error after coarse time delay calibration. The impact of the precise time delay error is more significant than the phase error when the sampling rate is low. This means that if the sampling rate can be increased, the precise time delay error will be reduced because the time delay error can be calibrated at the coarse time delay calibration step. The higher sampling rate will make the coarse time delay calibration more accurate. However, the increased sampling rate will also increase the number of samples to be processed, which will increase the computational complexity of the calibration process.

The proposed method generates the error matrix by applying a matched filter with the sliding window to the error signal. The computational complexity may be a problem when the number of TRMs is large. In this case, the sub-band concept can be applied to reduce the computational complexity while maintaining the frequency-dependent error monitoring capability as reported in [19].

The proposed method is compared with the sub-band approach as shown in Figure 15. The number of the sub-band is selected as 20 for the 25 MHz bandwidth. The figure shows that the sub-band matched filtering follows the tendency of the frequency-dependent error; however, the values are not as accurate as those of the proposed method. In the case that the frequency-dependent errors are not changing dramatically, the sub-band approach can be applied to reduce the computational complexity. However, at the frequency band where the frequency-dependent errors are changing rapidly, the sub-band approach cannot be applied because the errors are not accurately measured. By increasing the sliding step size to be the same as the window size, the proposed method can be converted to the sub-band approach. Based on the system parameters and signal processing capability, the window size and frequency resolution can be adjusted.

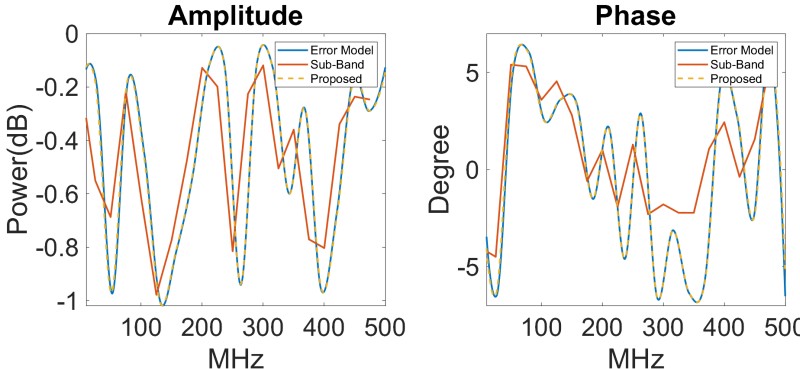

**Figure 15.** Calibration result comparison between the proposed method and sub-band calibration concept. The sub-band matched filtering follows the tendency of the frequency-dependent error; however, the values are not as accurate as the proposed method.

To analyze the effects of window size and sliding step size on the calibration performance, the RMSE of the amplitude and phase from the proposed method is analyzed with a different window size and overlapping ratio. The RMSE is calculated by comparing the calibrated error values from the proposed method with the randomly generated error values. The simulation is repeated 10 times with different random error values. The results are presented in Figure 16. The figure shows that the RMSE of the amplitude and phase is reduced as the window size and larger overlapping ratio. As the window size and sliding step size increases, the number of matched filter calculations decreases. Therefore, higher complexity can be traded off with higher calibration performance.

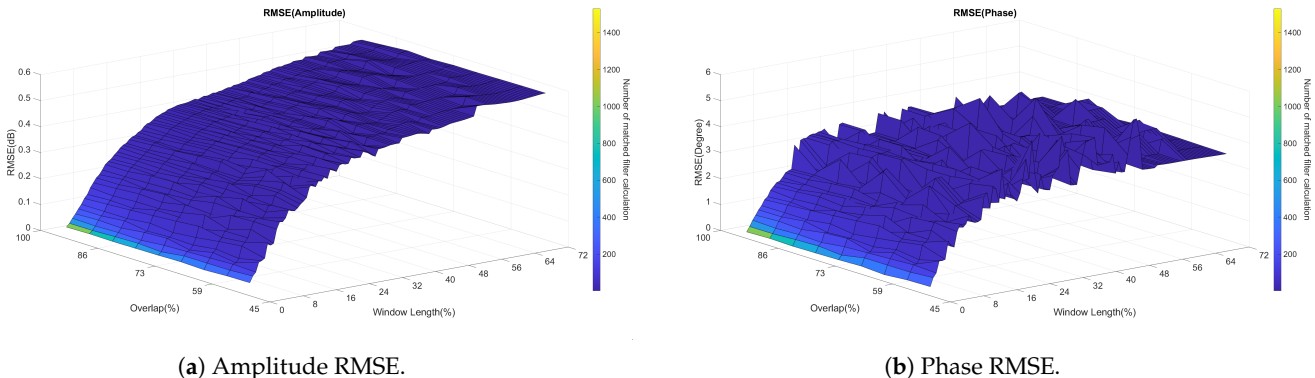

(**a**) Amplitude RMSE.

(**b**) Phase RMSE.

**Figure 16.** Trend of the RMSE of the proposed method with different window size and overlapping ratio. The colors in the surface indicate the number of matched filters, which can be considered the relative computational complexity. The root-mean-squared-error values are obtained after 10 iterations with randomized type 2 error.

## 5. Conclusions

The multifunctional and high-performance radar system requires the wideband signal to utilize the high-range resolution. The beamforming of the wideband signal requires the precise time delay calibration of TTD elements and the frequency-dependent amplitude and phase error from the TRM. As known in the literature, the wideband beamforming requires precise time delay control for each TRM.

This paper presents a new calibration method for the wideband LFM beamforming radar system. The proposed method can calibrate the frequency-dependent amplitude and phase error with precise time delay error without any additional hardware. By applying the stretched processing to the calibration signal, the proposed method can measure and calibrate the LFM signal with frequency-dependent amplitude and phase error with precise time delay error. The error measurement accuracy and its impact on the beamforming result

are analyzed. The proposed method is compared with true error values and conventional calibration results, and its calibration performance is verified by the simulation results. The accuracy of the proposed method is determined by the following factors: the sampling rate, and the window size and sliding step size of the stretched processing.

The proposed method can be applied with and without the orthogonal code-based calibration method. This is because the main process of the proposed method enables frequency-dependent monitoring in the sliding window algorithm, which can be applied to any LFM-based radar transceivers. Also, other types of transceivers can adopt the proposed method by generating the LFM signal for calibration purposes. The proposed method can be extended to the optimization of the parameter selection of the stretched processing for reducing the computational complexity.

**Author Contributions:** Conceptualization, N.K. and B.K.; methodology, B.K. and H.K.; software, H.K.; validation, H.K. and S.H.; formal analysis, S.H.; investigation, B.K., H.K, J.K. (Jinha Kim) and J.C.; resources, B.K. and H.K.; data curation, B.K. and H.K.; writing—original draft preparation, H.K.; writing—review and editing, B.K. and H.K.; visualization, B.K., H.K. and J.K. (Jinwoo Kim); supervision, B.K. and N.K.; project administration, S.H. and N.K.; funding acquisition, S.H. and N.K. All authors have read and agreed to the published version of the manuscript.

**Funding:** This work was supported by a grant-in-aid of HANWHA SYSTEMS.

**Conflicts of Interest:** The authors declare no conflict of interest.

## Abbreviations

The following abbreviations are used in this manuscript:

| | |
|---|---|
| LFM | Linear Frequency Modulation |
| TRM | Transmit and Receive Module |
| TTD | True Time Delay |
| PN | Pseudorandom Noise |
| BW | Bandwidth |

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
