# Peer review of "Calibration of Wideband LFM Radars Based on Sliding Window Algorithm"

_electronics, doi:10.3390/electronics12173564_

Round 1

Reviewer 1 Report

It is a little difficult to read. 

Reviewer 2 Report

None

Reviewer 3 Report

This paper described new calibaration method based on sliding window algorithm. The paper was well written and the effectiveness of the proposed technique was well revealed through experiments. However, there are a few things that need to be revised for the publication of the paper.

1. The motivation for using sliding window algorithm and its effects need to be described in more detail in the introduction section.

2. How can I determine the appropriate window size and sliding step size in the real world? The methodology for this part needs to be mentioned in this paper.

3. In the Results section, the results of all the experiments are illustrated by figures, but these need to be quantified and organized into a table.

None

Round 2

Reviewer 1 Report

see attached

please check English again!
